# Effect of a 3.5% NaCl−10% HCl Corrosive Environment on the Fatigue Behavior of Hot Rolled Aluminum 5083-H111

**DOI:** 10.3390/ma16144996

**Published:** 2023-07-14

**Authors:** Masoud M. M. Elhasslouk, İsmail Esen, Hayrettin Ahlatcı, Bengu Akın

**Affiliations:** 1Mechanical Engineering Department, Karabuk University, Karabuk 78050, Turkey; masoudhaslook2016@gmail.com; 2Metallurgical and Materials Engineering Department, Karabuk University, Karabuk 78050, Turkey; hahlatci@karabuk.edu.tr (H.A.); benguakin_@hotmail.com (B.A.)

**Keywords:** H11, Al 5083, corrosion, fatigue life

## Abstract

This study deals with the microstructure of rolled Al5083-H111 materials, their hardness, corrosion in different solutions, and rotary bending fatigue properties of non-corroded and corroded samples in different solutions. This study is the first to report the fatigue behavior of corroded samples in different aggressive corrosion environments of Al5083. The microstructure of the Al5083-H111 material is in the form of grains oriented towards the rolling direction and it consists of binary Al-Mg, Al-Mn, and Mg-Si; ternary Al-Mg-Si; and quaternary Al-Mn-Fe-Si and Al-Cr-Mn precipitated randomly at the grain boundary. The Brinell hardness of the Al5083-H111 material is 68.67 HB. According to the results of the immersion corrosion, while the sample was more resistant to corrosion in a 3.5% NaCl environment, it showed a less resistant behavior in a 3.5% NaCl + 10% HCl environment. As a result of the fatigue test, it was observed that the sample that did not undergo corrosion showed a higher fatigue life than the samples that were exposed to corrosion. The fatigue rate of the 3.5% NaCl corrosion sample was 3.5 times lower than the fatigue rate of the 3.5% NaCl + 10% HCl corrosion sample.

## 1. Introduction

The lightweight, corrosion resistance, and mechanical properties of aluminum make it indispensable in technological use [1]. Aluminum and its alloys possess versatile properties that make them suitable for use in construction and engineering. Consequently, their utilization in the industrial domain has experienced a significant surge in recent times. The significance of aluminum has been amplified in the computer, aerospace, and automotive sectors, owing to its exceptional strength, lightweight quality, and high electrical and thermal conductivity in a majority of its alloys. The behavior of this material, which exhibits a wide variety of combinations of properties, has been the subject of many studies, and research on this subject is still ongoing [2,3,4]. Despite all of the good properties of aluminum-based materials, the use of this metal is still limited because of its low strength values. In particular, the low strength values of pure aluminum (30 MPa yield strength, 186 MPa tensile strength) are increased through alloying (250 MPa yield strength and 434 MPa tensile strength) [5]. The fatigue performance of high-strength aluminum alloys is rather disappointing. There is no corresponding improvement in fatigue quality; the increase in static tensile properties is achieved through the addition of other alloying elements, including using Mg as the main alloying element [5]. Aluminum alloys are divided into two according to the method for forming the product. These are wrought aluminum alloys and cast aluminum alloys [6]. Alloys of aluminum denoted with the numerical digit “5” are primarily composed of magnesium as the principal alloying constituent. The majority of wrought alloys in this category typically have a magnesium content of less than 5% [7]. Table 1 displays the customary chemical composition of this alloy.

Aluminum and its alloys are a significant group of engineering materials that are both lightweight and resistant to corrosion. The density of pure aluminum is 2.70 g/cm^3^, which allows for certain aluminum alloys to exhibit superior strength-to-weight ratios compared with high-strength steels [7]. Al-Mg alloys belonging to the 5000 series, wherein Mg serves as the primary alloying constituent, are categorized as non-heat treatable alloys that exhibit commendable resistance to corrosion and weldability [9]. The commercial wrought alloy known as aluminum alloy 5083 derives its strength from the combined effects of magnesium solute hardening and strain hardening [10]. The application of 5083 and other 5xxx alloys in shipbuilding is attributed to their elevated specific strength, weldability, and commendable resistance to corrosion in marine settings [9,10,11]. The corrosion resistance exhibited by aluminum alloys is ascribed to the autonomous generation of a slender, dense, and adhesive layer of aluminum oxide on the external surface upon exposure to air or water. The hydrated layer of aluminum oxide has the tendency to experience dissolution in certain chemical solutions, such as strong acid or alkaline solutions. The corrosion phenomenon known as pitting corrosion can arise due to the localized attack of materials when the passive layer is compromised in chloride-rich environments, such as those found in NaCl solutions or in sea water [12,13,14,15]. The formation and expansion of corrosion pits in hostile surroundings, as well as the onset and progression of fatigue fractures, are expedited by the presence of precipitates, secondary-phase particles, and voids [16,17,18,19]. Previous research has confirmed the detrimental effect of immersion in a sodium chloride solution on the fatigue characteristics. The present investigation centered on examining the fatigue and corrosion−fatigue characteristics of welds made from 5083-H111 aluminum alloys [7].

The corrosion fatigue behavior of a conventional counterpart, 5083(H111), and a nanocrystalline ultrafine grain (UFG) Al-Mg-based alloy was investigated by Sharma et al. [20]. The experimental findings indicated that the ultrafine-grained (UFG) alloy exhibited a higher level of fatigue resistance in comparison with the conventional 5803 alloys. The findings of Sharma et al. indicated that upon microscopic analysis of standard 5083 [20] fatigue samples subjected to corrosion in the 3.5% NaCl solution, the emergence of a crack leading to eventual corrosion fatigue generally originated from the β-phase inclusions or the corrosion pit, which acted as stress concentrators initiating mechanical failure. The fatigue crack growth behavior of the aluminum alloy 5083-H131 was systematically examined by Holtz et al. [21], with a focus on the degree of sensitization resulting from aging at 448 K (175 C). The authors observed that a material’s risk of corrosion−fatigue failure was significantly elevated when its ASTM G-67 mass loss value surpassed 30 mg/cm^2^, particularly when subjected to load ratios ranging from 0.8 to 0.9 in the presence of corrosive media.

Considering the usage area of the 5083 alloy, which is one of the aluminum 5XXX series, many researchers have supported the development of its mechanical properties due to the corrosion environment it is exposed to. However, when we look at the literature, although a significant amount of damage in metal alloys is caused by fatigue behavior, there are almost no studies on this subject [21]. The aim of this study was to investigate the corrosion behavior and fatigue behavior of 5083-H111 after corrosion in solutions containing 3.5% NaCl and 3.5% NaCl + 10% HCl at 24, 48, and 72 h intervals, both in uncorroded and corroded conditions. Thus, we aimed at taking the first step in experimental studies considering the strength of materials that will contribute to the corrosion and corrosion fatigue behavior of Al5083 in different corrosion environments that can be used in appropriate areas.

## 2. Materials and Methods

In this study, the hot-rolled Al-5083-H111 material with EN 485, EN 515, and EN 573-3 [1] production standards obtained from SEYKOÇ ALUMINUM was used. The microstructure, hardness, and corrosion fatigue properties of the supplied materials were investigated.

The elemental composition of the hot-rolled Al-5083-H111 sample was determined at KBU MARGEM using the X-ray fluorescence procedure (XRF) of the brand Rigaku ZSX Primus II and the X-ray characterization of the elements and percent by weight.

The Al5083 sample’s current compositions and phases were determined by obtaining X-ray diffraction (XRD) profiles using a Rigaku Ultima IV device. The scanning range was 10–90° and the scanning speed was 3°/min.

The hot rolled Al-5083-H111 sample was first cut with a water-cooled band saw in 10 × 20 × 10 mm size for microstructure characterization. The sample, whose cutting process was completed, was sanded in a Mikrotest brand automatic sanding and polishing device. Sanding was done with sandpaper coated with 320, 400, 600, 800, 1000, and 2500 grid SiC particles. After sanding, the polishing process was completed with a 3 μm Al_2_O_3_ liquid solution. Keller etching with 2 mL of hydrogen fluoride, 3 mL of hydrochloric acid, 5 mL of nitric acid, and 190 mL of distilled water was used for pickling. A Carl Zeiss optical microscope was used for the microstructure examination. For detailed investigations, Carl Zeiss Ultra Plus Gemini brand SEM and for phase morphologies the EDX associated with the SEM device were examined.

Hardness tests of the Al5083 samples were determined with an HB-3000B brand Brinell hardness tester at KBU Metallurgical and Materials Engineering Laboratory. The hardness test was repeated five times with 187.5 force and 2.5 mm diameter steel balls.

The surfaces of cube-shaped specimens with dimensions of about 10 mm, 11 mm, and 12 mm prepared for immersion corrosion tests were first cleaned in an ultrasonic cleaner. The surface area of each sample was calculated one by one and the weight measurements were made with a Precisa brand precision balance. The 3.5% NaCl solution and 3.5% NaCl + 10% HCl solution were used for the immersion corrosion. The corrosion solutions were placed in jars and the samples were left suspended. The immersed samples were removed from the solution at intervals of 24, 48, and 72 h; their surfaces were cleaned; and their weights were measured and recorded. The corrosion products formed on the surface of the samples were removed at an interval of every hour by keeping them in a chromic acid solution prepared in pure water at a rate of 180 g/L for 10 min. The samples were then cleaned with ethyl alcohol in the ultrasonic vibration device and left in the solution again. As a result of the immersion corrosion test, the decrease in the weight of the sample per hour and the total corrosion rates were calculated. In addition, fatigue samples were kept in these corrosion solutions for 24, 48, and 72 h. The results were repeated on at least two other samples. At the end of the 72 h immersion test, the SEM images of the corroded surfaces of each sample were taken and the corrosion mechanisms were examined in detail.

Fatigue tests of the hot-rolled Al-5083-H111 specimens were performed on a rotating bending fatigue test machine under repeated bending stresses with respect to a continuously rotating neutral axis, given in Figure 1. The samples were created in accordance with the technical drawing given in Figure 2. For fatigue test specimens with a diameter of 6.80 mm, vise heads suitable for a diameter range of 6.5–7 mm were used (Figure 3). Fatigue tests were performed on hot-rolled Al-5083-H111 specimens in pure and corroded conditions after 24, 48, and 72 h. Fatigue lifetimes were determined by at least two samples under constant load (48.5 kg) and constant speed (25 Hz). The fracture surfaces were visualized by SEM in their pure state after the rotational bending fatigue test and the corrosion state after 24 and 72 h (3.5% NaCl and 3.5% NaCl and 3.5% NaCl + 10% HCl). The fractured surface mechanisms were studied in detail.

## 3. Results and Discussion

### 3.1. Microstructural Characterisation

The average chemical compositions of hot-rolled Al-5083-H111 used in the study are given in Table 2. Figure 4a shows the 20× magnification optical microscope, Figure 4b shows the 50× magnification optical microscope, Figure 4c shows the 1 K× magnification SEM image, and Figure 4d shows the 5 K× magnification SEM image results. In the optical microstructure image of the 5083 Al material, blackish and gray-colored particles were scattered on the matrix. It can be seen that the grains were in the form oriented toward the rolling direction. It can be seen that intermetallic phases oriented in the rolling direction precipitated at the grain boundary and were not uniformly distributed.

The microstructure images were characterized based on the SEM image in Figure 5 and the EDX results from Figure 5 (Table 3) and the XRD spectra in Figure 6. In Figure 6, the XRD spectra of the Al5083 sample are given. Al_6_Mn was the most common phase peak. The XRD peaks of the Al5083 alloy started at 20° for the Al_3_Mg_2_, Al_6_Mn, and Al(Mn, Fe)Si phases. The Al_3_Mg_2_, Al_12_Mg_17_, Al_6_Mn, and Mg_2_Si phases were determined at the peaks seen at 38.5°. The Al_6_Mn, Mg_2_Si, and AlMgSi phases were seen at 44.5°. The Al_12_Mg_17_, Al_3_Mg_2_, and Al_6_Mn at 65°; Al_3_Mg_2_, Al_12_Mg_17_, and Al_6_Mn at 78.5°; and the Al_18_(Cr,Mn)_2_Mg_3_ and Al_12_Mg_17_ at 80° were detected. Finally, the XRD peaks of the Al_3_Mg_2_, Al_6_Mn, and Al(Mn, Fe)Si phases ended at 82°.

According to the XRD results, binary Al-Mg, Al-Mn, and Mg-Si;ternary Al-Mg-Si and Al-Mn-Fe; and quaternary Al-Mg-Mn-Si and Al-Cr-Mn-Mg intermetallics were detected. The composition of the alloy comprised a solid solution that was rich in aluminum and an alpha-aluminum matrix. The particles present in the alloy exhibited a bright and dark appearance and were primarily oriented along the direction of rolling. The presence of 94.81% Al and 5.19% Mg in Region 1, shown in Table 3, indicates the matrix. In region 2, the presence of 63.76% Si, 33.29% Al, and 2.06% Mg elements, various intermetallic phases such as Al-Mg (almost the same color as the Al matrix), Al-Mg-Si (bright surrounding gray), and Mg_2_Si indicates that [22]. The existence of similar types of intermetallics has been reported by previous researchers [23,24]. Region 3 also contained 93.34% Al, 4.62% Mg, 0.92% Si, 0.60% Fe, 0.35% Mn, and 0.16% Cr. The Al_6_(Mn-Fe-Cr) phase, which has also been described in some studies in the literature [25,26,27], is believed to be the bright phases appearing in the SEM images in Figure 5 region 3. Some studies show that a quaternary α-Al(Mn, Fe)Si phase was formed with the Al_8_FeMnSi_2_ or Al_12_(Fe, Mn)_3_Si composition in the Al 5xxx alloys [28,29,30,31]. Therefore, region 3 could be a-Al(Mn, Fe)Si. Region 4 shows that there could be a Cr-rich ε-Al_18_(Cr, Mn)_2_Mg_3_ phase without the Fe and Mg_2_Si phase due to its dark color [24,32]. The authors of the study, L. Tan and T.R. Allen, identified four distinct types of precipitates through the use of energy dispersive spectroscopy (EDS). These precipitates were classified as large white Al-(Fe, Si, Mn, Cr), medium black Mg_2_Si, small white Al_13_Fe_4_, and dense ultrafine Al_6_Mn. According to their statement, SEM imaging did not capture the β-phase as it was of a diminutive scale, measured in the nanometer or submicron range [33].

### 3.2. Hardness Test Results

The hardness result of the hot-rolled Al-5083-H111 sample was 68.67 ± 1.84 HB/2.5/187.5/10. Here, 68.67 represents the Brinell hardness value, 2.5 mm steel ball diameter, 187.5 kg applied load, and 10 s time.

### 3.3. Immersion Corrosion Test Results

The literature study examined the corrosion resistance of 5083 and discussed the following aspects. The 5083-aluminum alloy belongs to the 5xxx series of aluminum alloys and is characterized by the presence of magnesium as the primary alloying element, which is added to enhance its corrosion resistance properties. The susceptibility of pitting attacks and intergranular corrosion in the 5083 aluminium alloy, particularly in those with magnesium levels exceeding 3% by weight, can be attributed to the intricate precipitation occurring at the grain boundaries [34,35]. Immersion experiments were evaluated in terms of weight loss changes over time, given in Figure 7, by measuring the weight losses in 3.5% NaCl and 3.5% NaCl + 10% HCl solutions at room temperature at 24, 48, and 72 h time intervals. According to Figure 7, the weight loss after 24 h in the 3.5% NaCl solution was 0.000005 mg/dm^2^, while the solution containing 3.5% NaCl + 10% HCl was calculated as 0.000017 mg/dm^2^. According to the results of the immersion corrosion, while the sample was more resistant to corrosion in a 3.5% NaCl environment, it showed less resistant behavior in a 3.5% NaCl + 10% HCl environment. The reason for this is thought to be intergranular corrosion in the material in the 3.5% NaCl + 10% HCl corrosion environment. After 72 h, the highest weight loss occurred with 0.000041 mg/dm^2^ in the 3.5% NaCl + 10% HCl solution, while the lowest weight loss was calculated with 0.000014 mg/dm^2^ in the 3.5% NaCl environment.

The mdd (milligrams per square decimeter per day) calculation, which was used in the determination of the corrosion rate by using the weight losses of 24, 48, and 72 h, was made and the mdd values are given in Figure 8 in mg/dm^2^·day. According to the corrosion rate data given in Figure 8, the sample in the 3.5% NaCl environment showed the lowest corrosion rate with a value of 0.0000046 mg/(dm^2^·day). The highest corrosion rate was determined with 0.000017 mg/(dm^2^·day) in a 3.5% NaCl + 10% HCl sample. As seen in Figure 8, the sample showed a more stable corrosion behavior after 24 h in a 3.5% NaCl environment. However, the same situation was not valid for a 3.5% NaCl + 10% HCl environment.

Figure 9 shows the corrosion rate values of the Al5083 samples at the end of 72 h. At the end of 72 h, the corrosion rate of the sample in the 3.5% NaCl solution was 0.000005 mg/dm^2^·day, while the corrosion rate of Al5083 in the 3.5% NaCl + 10% HCl solution was 0.000014 mg/dm^2^·day.

In Figure 10, the SEM images of the Al-5083 sample corroded in the 3.5% NaCl solution at different magnifications are given. Figure 11 shows the 1K× magnification SEM image of the Al-5083 sample corroded in the 3.5% NaCl solution, while Table 4 shows the EDX analysis results from Figure 12. The SEM image shows that corrosion occurred in the form of homogeneous and stratified separation.

The post-corrosion scanning electron microscopy (SEM) and energy-dispersive X-ray spectroscopy (EDS) analysis of AA5083 revealed the presence of five intermetallic phases that were observed to be non-uniformly dispersed within the aluminum matrix. As per the literature, the intermetallic phases primarily consisted of Al_6_(Fe, Mn), Al_6_(Mn, Fe, Cr), AlMg, Mg_2_Si, and Al(Si, Mg) [36]. According to the literature, Al_6_(Mn, Fe, Cr) precipitates exhibited a greater cathodic behavior compared with the aluminum matrix [37]. Consequently, the aforementioned precipitates underwent a transformation into enduring cathodes through the process of oxygen reduction to hydroxide ions. The local rise in pH wsa a consequence of the dissolution of the oxide layer surrounding the precipitates. Upon the dissolution of this layer, the heightened alkalinity in the vicinity led to a vigorous assault on the matrix [37].

Al, Mn, and Fe elements detected in regions 3 and 5, shown in Table 4, indicate the Al_6_(Fe, Mn) phase. Prior research indicates that the Al_6_(Fe, Mn) phase exhibits a greater noble potential in comparison with aluminium [38]. Consequently, cathodic reactions take place within the alloy matrix [38], leading to the formation of pits. The process of oxygen reduction in the cathodic reaction generates hydroxyl anions, which facilitate the disruption of the oxide layer surrounding Fe-containing particles, thereby promoting the formation of pits [39]. On the other hand, it has been reported that the Mg–Si phase (Mg_2_Si) seen in regions 2 and 6 has similar corrosion potential values compared with the surrounding aluminum matrix [39] or a lower corrosion potential compared with the aluminum matrix [40].

In Figure 12, SEM images of the Al-5083 sample corroded in the 3.5% NaCl + 10% HCl solution at different magnifications are given. Figure 13 shows the 1KX magnification SEM image of the Al-5083 sample corroded in the 3.5% NaCl + 10% HCl solution, while Table 5 shows the EDX analysis results from Figure 13. As can be seen in Table 5, the same intermetallics were found in the post-corrosion SEM and EDX results of AA5083 in the 3.5% NaCl + 10% HCl solution, as well as the post-corrosion EDX results in the 3.5% NaCl solution. SEM images of the 3.5% NaCl + 10% HCl sample, in which the pitting intensified at the grain boundaries and the cavitation increased, also support the decrease in corrosion resistance. Looking at the SEM image, it is observed that shallow and deep cavities were formed. Crater formation is observed at the points where the cavities are sharper and the intermetallics are lifted compared with the 3.5% NaCl corrosion sample. This caused faster corrosion. The addition of 10% HCl increased the acidity and excess Cl ions in the structure increased cavitation.

According to EDX analysis, it can be seen that the oxygen ratio of the 3.5% NaCl + 10% HCl corrosion sample was higher than that of the 3.5% NaCl corrosion sample. We also attributed the worsening of the corrosion potential of the 3.5% NaCl + 10% HCl corrosion sample to the oxygen ratio in the structure. The higher the oxygen ratio, the higher the oxidation. It can clearly be seen that oxidation increased corrosion. The scanning electron microscopy (SEM) image revealed that the areas exhibiting corrosion were primarily located along the grain boundaries and in close proximity to said boundaries. The rationale behind this phenomenon was attributed to the precipitation of phases along the grain boundaries, which subsequently elevated the susceptibility to corrosion. Corrosion resistance decreased in the presence of the β-phase and Mg_2_Si. Beta phase, Mg_2_Si, and Al (Fe, Cr, and Mn) precipitates detected by XRD are known to cause localized corrosion [20].

### 3.4. Rotary Bending Fatigue Test Results

The rotary bending fatigue test results of the hot-rolled Al-5083-H111 sample are given in Figure 14. When the fatigue results were examined in general, it was seen that the non-corrosive sample showed better fatigue life than the samples exposed to corrosion. However, as can be seen from the corrosion test results of the sample, the 3.5% NaCl corrosion fatigue sample also showed better results in the fatigue tests compared with the 3.5% NaCl + 10% HCl corrosion fatigue sample. While the fatigue life of the uncorroded sample was 11,763,207.5 cycle, it was measured as 11,753,950 for the 3.5% NaCl sample corroded for 24 h, and 11,731,860 cycle for the 3.5% NaCl + 10% HCl sample. The fatigue lives of the samples, which were corroded for 48 h, was found to be 11,746,375 and 11,695,280 cycles, respectively. Furthermore, 11,737,060 cycles of the 3.5% NaCl sample and 11,673,025 cycles of the 3.5% NaCl + 10% HCl sample remaining in a corrosive environment for 72 h were calculated. The fact that the fatigue life of the sample corroded in the 3.5% NaCl solution did not decrease sufficiently was due to less pitting during corrosion. The stable difference deteriorated after 48 h and the fatigue life of the sample exposed to corrosion in a 3.5% NaCl + 10% HCl environment decreased the most in the following hours compared with the other samples.

The fatigue rate of the Al5083/3.5% NaCl corrosion fatigue specimen was calculated as 0.74 N/(h·L). Here, N is the fatigue life number, the applied load is given in as L in Newtons, and h is the corrosion exposure time in hours. In the Al5083/3.5% NaCl + 10% HCl corrosion fatigue sample, the fatigue rate increased by 2.64 N/(h·L), according to the fatigue life results of the Al5083/3.5% NaCl sample. Fatigue rate increased by 257% in the HCl sample compared with the Al5083 3.5% NaCl corrosion fatigue sample.

In Figure 15, Figure 16 and Figure 17, the cracked surface SEM examinations of the uncorroded samples at different magnifications are given after the fatigue tests of the corroded samples in the 3.5% NaCl and 3.5% NaCl + 10% HCl solutions. Supporting the corrosion and corrosion fatigue test results, when the SEM images were examined, it was seen that the fatigue strength of the uncorroded sample was better.

Chlistovsky et al. [41] demonstrated in their work that the fatigue life of the 7075-T651 alloy experienced a notable reduction when exposed to a corrosive medium of 3.5% NaCl. The decrease in size was ascribed to the commencement of fractures due to the stress concentration resulting from the formation of pits, in addition to a blend of anodic dissolution and hydrogen embrittlement at the highest point of the fracture. A study conducted by [19] examined the corrosion fatigue characteristics of aluminum 2024-T3. The research conducted by the authors demonstrated that the formation of fatigue cracks occurred as a result of nucleation from one or two sizable pits that were visibly present on the surface.

The fact that the pitting of the 3.5% NaCl + 10% HCl corrosion fatigue sample was higher and larger in the SEM image compared with the other samples supports that the fatigue strength of the sample was badly affected in this solution. Another factor is that the fatigue crack progressed over the entire surface. The process of corrosion pit nucleation and growth is expedited in aggressive environments owing to the existence of precipitates, second-phase particles, pores, and grain boundaries within the matrix. Furthermore, these characteristics facilitate the onset and progression of fatigue fractures [7].

The Mg_2_Si phase detected in the XRD results (Figure 6) has been stated in the literature to be very harmful to the mechanical properties of the alloy due to its uneven distribution and irregular shape [42]. The alloys Al_3_Mg_5_ and Mg_2_Si exhibited anodic behavior in relation to the alloy matrix 5083, thereby facilitating rapid localized corrosion via galvanic coupling. The impact of cathodic corrosion on the fatigue characteristics of conventional 5083 alloys is noteworthy, as the formation of significant pits in the vicinity of particles that are cathodic to the matrix can occur [20]. The pits formed after the corrosion test expanded and coalesced as a result of fatigue rotating bending loading, resulting in significant fracture propagation and expansion from the initial position. Previous studies on the same alloys [43,44,45] noticed this trend. A significant corrosion pit defect such as narrow, elliptical, and wide-deep pits (indicated by arrows in Figure 16 and Figure 17) occasionally developed in the investigated sample around an inner inclusion near the outer surface (Figure 16 and Figure 17).

## 4. Conclusions

The following conclusions were drawn regarding the microstructure, hardness, corrosion, and fatigue properties of the rolled Al5083-H111 material:It was seen that the microstructure of the Al5083-H111 material consisted of grains oriented towards the rolling direction. We noticed that intermetallic phases precipitated at the grain boundary and were not evenly distributed.The hardness result of the Al5083-H111 material was measured as 68.67 ± 1.84 HB.According to the results of the immersion corrosion, while the Al5083 sample was more resistant to corrosion in a 3.5% NaCl environment, it showed less resistant behavior in a 3.5% NaCl + 10% HCl environment. The reason for this is thought to be intergranular corrosion in the material in the 3.5% NaCl + 10% HCl corrosion environment. The sample showed a more stable corrosion behavior after 24 h in a 3.5% NaCl environment. However, the same situation was not valid for a 3.5% NaCl + 10% HCl environment. According to the corrosion rates at the end of 72 h, the presence of the 10% HCl solution increased the corrosion rate by 180%.According to the fatigue results, it was observed that the non-corrosive sample showed a better fatigue life than the samples exposed to corrosion. The 3.5% NaCl corrosion fatigue sample, on the other hand, provided better results in the fatigue tests compared with the 3.5% NaCl + 10%HCl corrosion fatigue sample. The stable difference deteriorated after 48 h and the fatigue life of the sample exposed to corrosion in a 3.5% NaCl + 10% HCl environment decreased the most in the following hours compared with the other samples. According to the fatigue rate results, the presence of 10% HCl solution in the corrosion electrolyte reduced the fatigue life by 257%.

## Figures and Tables

**Figure 1 materials-16-04996-f001:**
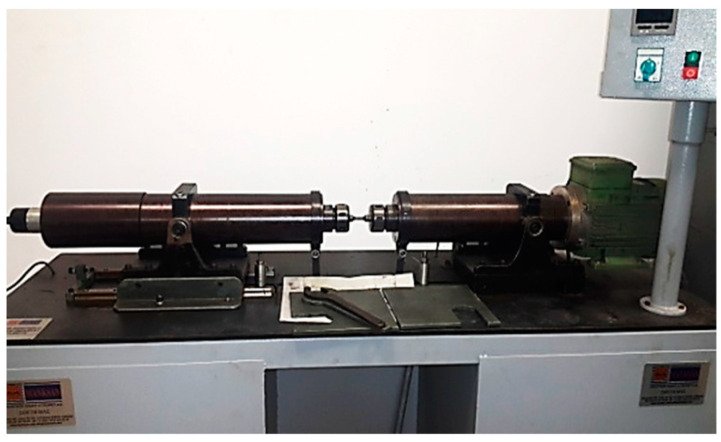
Rotating bending fatigue tester.

**Figure 2 materials-16-04996-f002:**
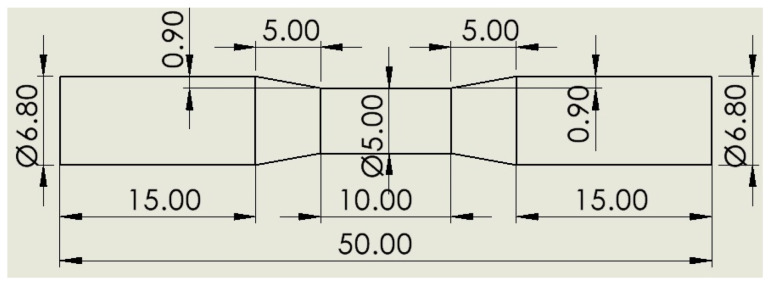
Schematic view and dimensions (in mm) of the technical drawing on the fatigue specimen.

**Figure 3 materials-16-04996-f003:**
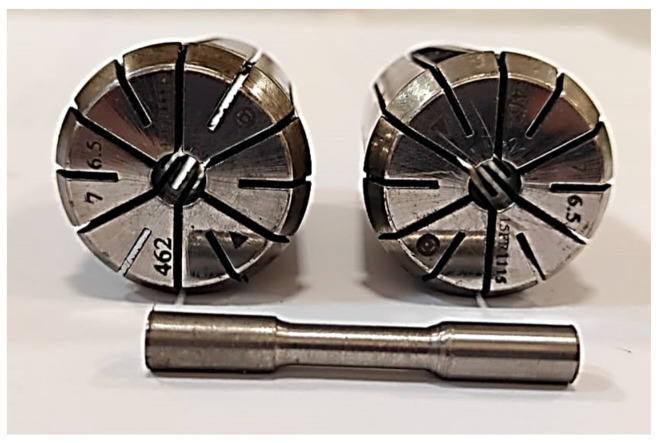
Specimen and clamp heads used in the rotating bending fatigue testing.

**Figure 4 materials-16-04996-f004:**
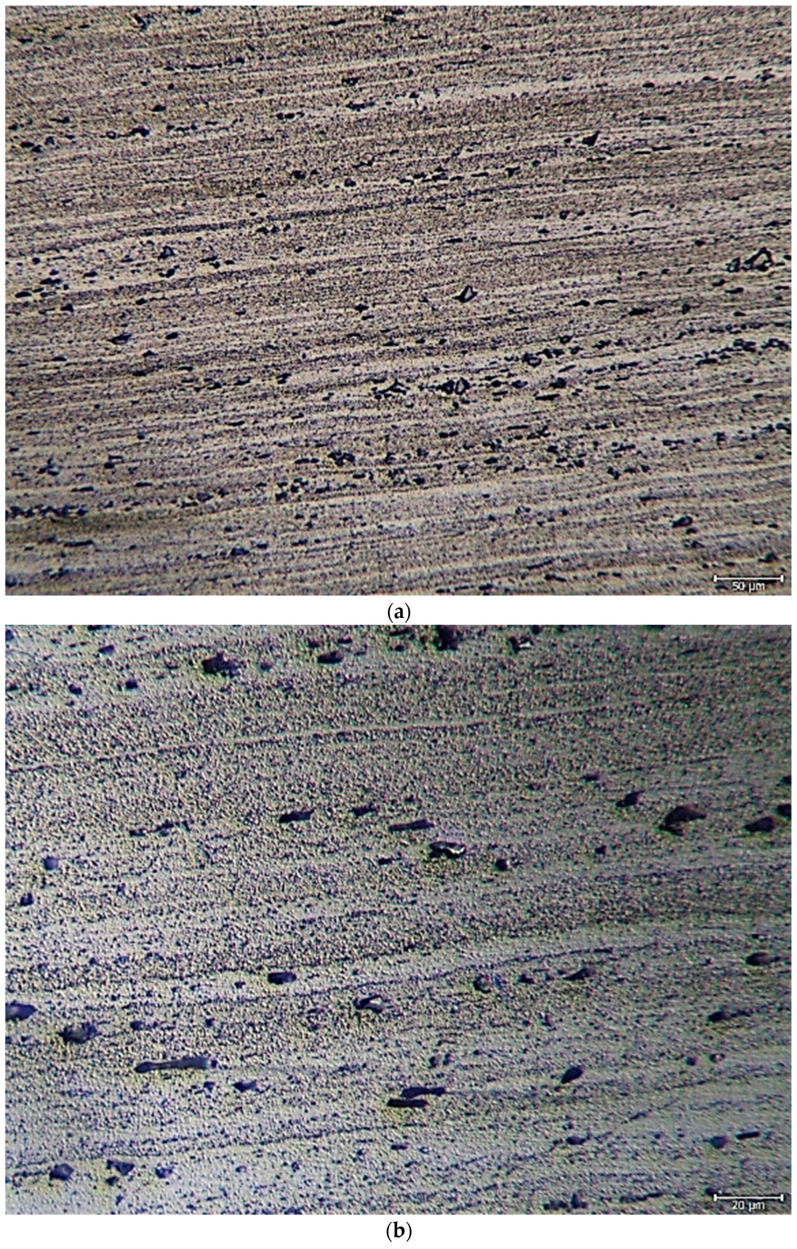
Hot-rolled Al-5083-H111: (**a**) 20× magnification optical microscope image, (**b**) 50× magnification optical microscope image, (**c**) 1 K× magnification SEM image, and (**d**) 5 K× magnification SEM image.

**Figure 5 materials-16-04996-f005:**
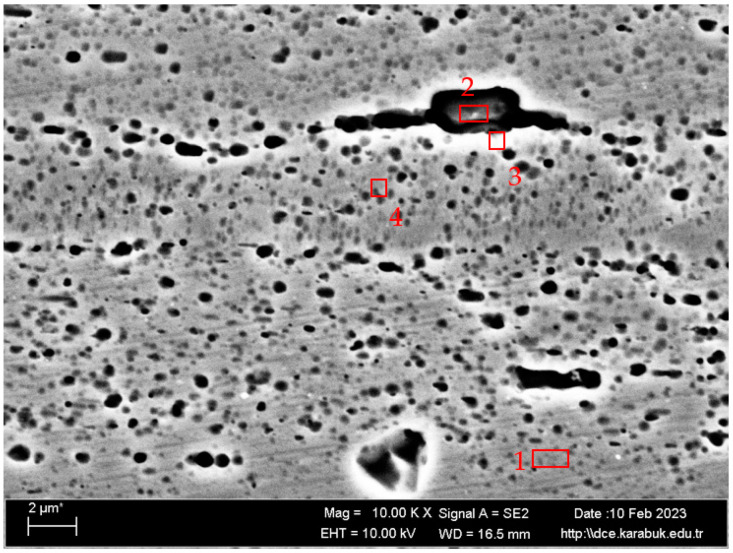
The 10 K× magnification SEM image of the hot-rolled Al-5083-H111 sample.

**Figure 6 materials-16-04996-f006:**
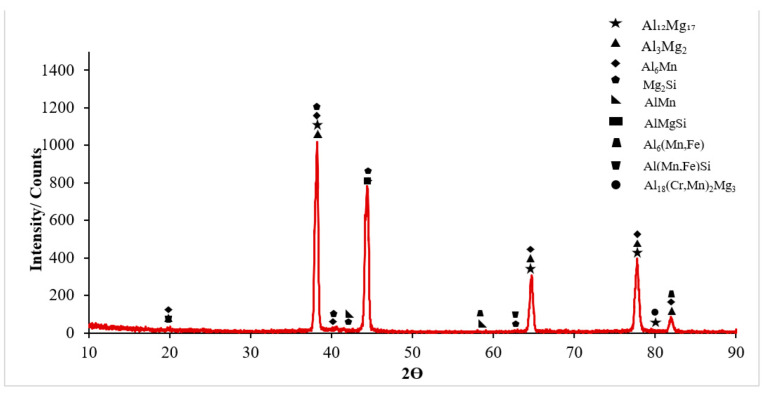
XRD result of the hot-rolled Al-5083-H111 sample.

**Figure 7 materials-16-04996-f007:**
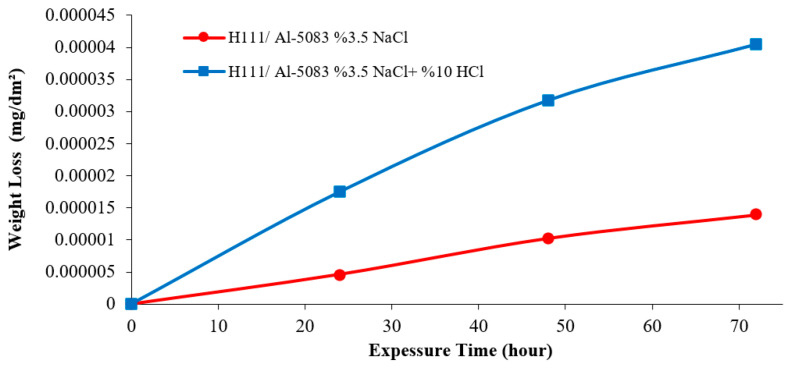
Immersion corrosion of the hot-rolled Al-5083-H111 sample as a result of weight loss.

**Figure 8 materials-16-04996-f008:**
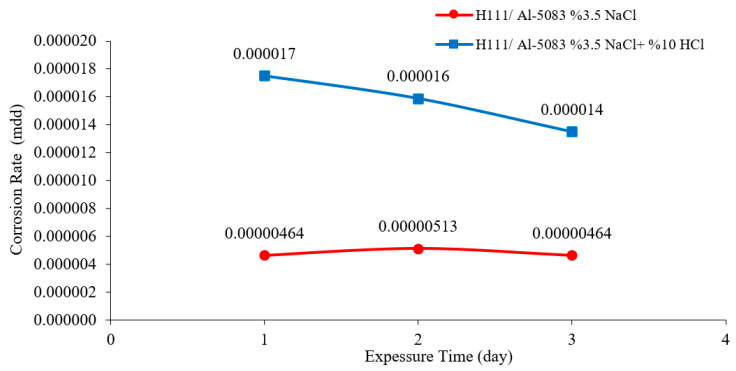
Results of corrosion rates for the hot-rolled Al-5083-H111 sample.

**Figure 9 materials-16-04996-f009:**
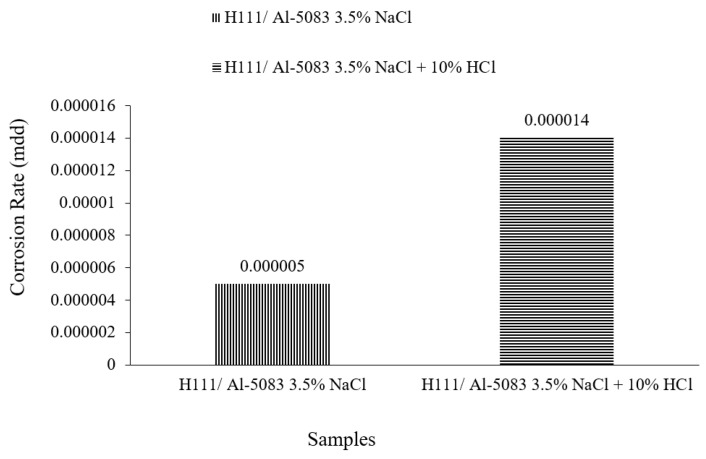
Corrosion rates of Al5083-H111 after 3 days.

**Figure 10 materials-16-04996-f010:**
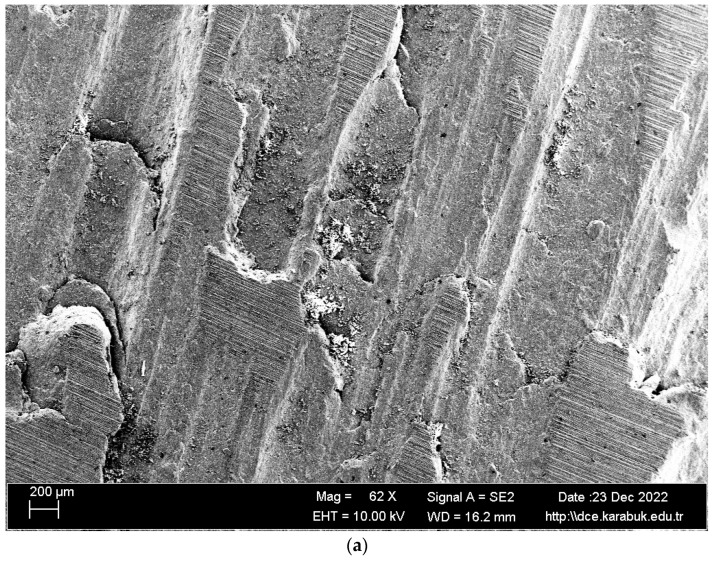
SEM image of Al-5083 sample after being corroded in the 3.5% NaCl solution: (**a**) 62× and (**b**) 500× magnification.

**Figure 11 materials-16-04996-f011:**
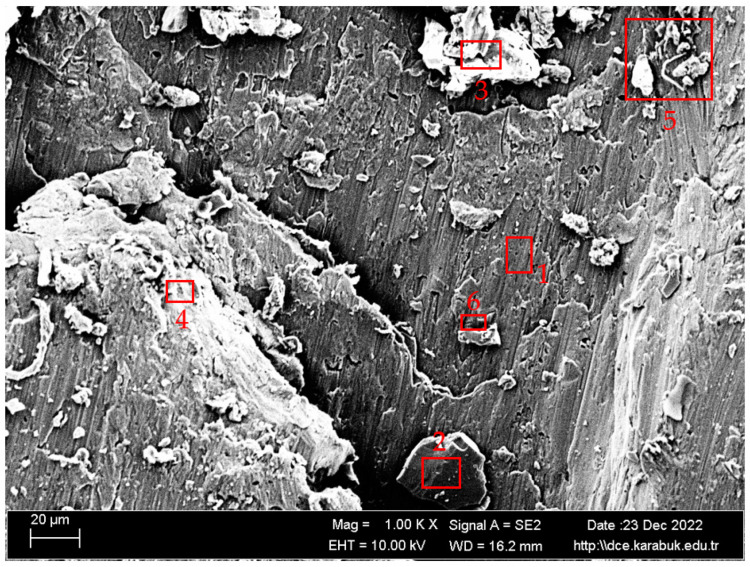
1K× magnification SEM image of the Al-5083 sample corroded in the 3.5% NaCl solution taken from region A in Figure 10.

**Figure 12 materials-16-04996-f012:**
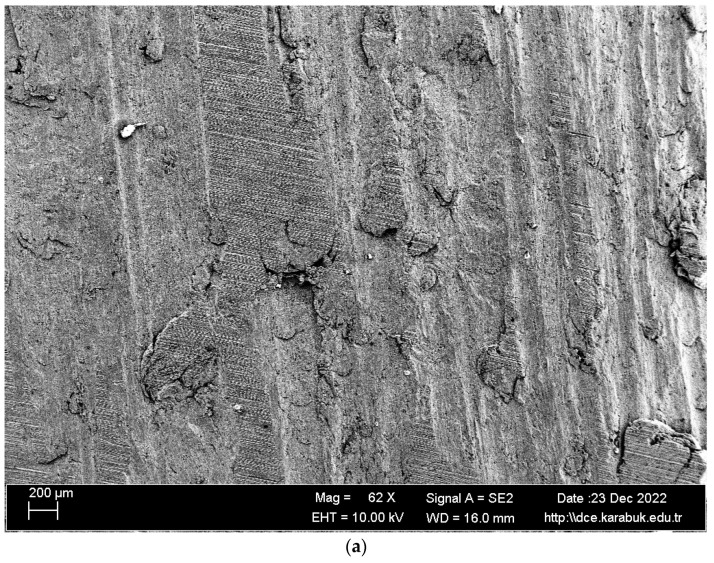
SEM image of the Al-5083 sample corroded in the 3.5% NaCl + 10% HCl solution: (**a**) 62× and (**b**) 500× magnification.

**Figure 13 materials-16-04996-f013:**
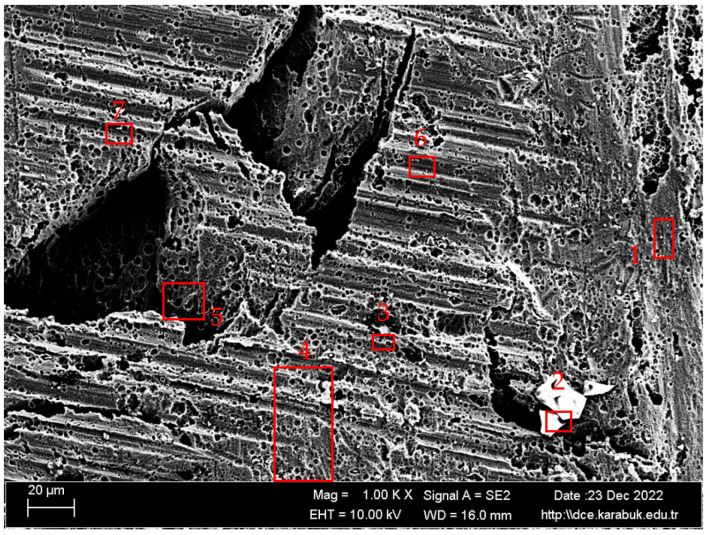
1K× magnification SEM image of the Al-5083 sample corroded in the 3.5% NaCl + 10% HCl solution taken from region B in Figure 12.

**Figure 14 materials-16-04996-f014:**
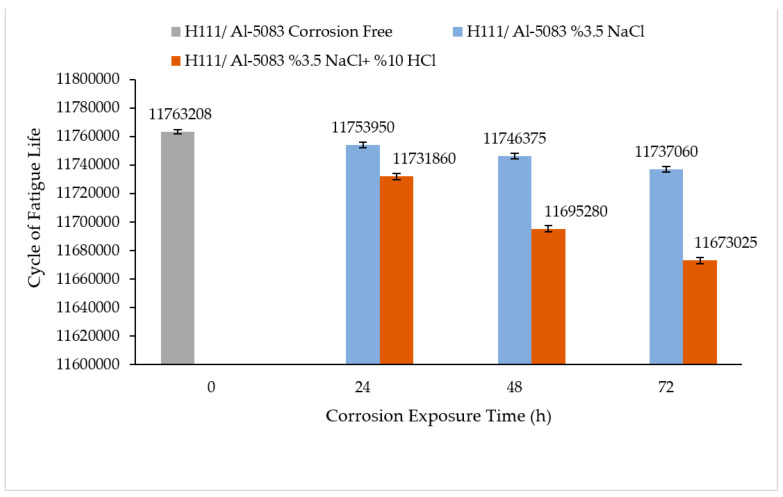
Rotating bending fatigue test results of the hot-rolled Al-5083-H111 sample before and after corrosion.

**Figure 15 materials-16-04996-f015:**
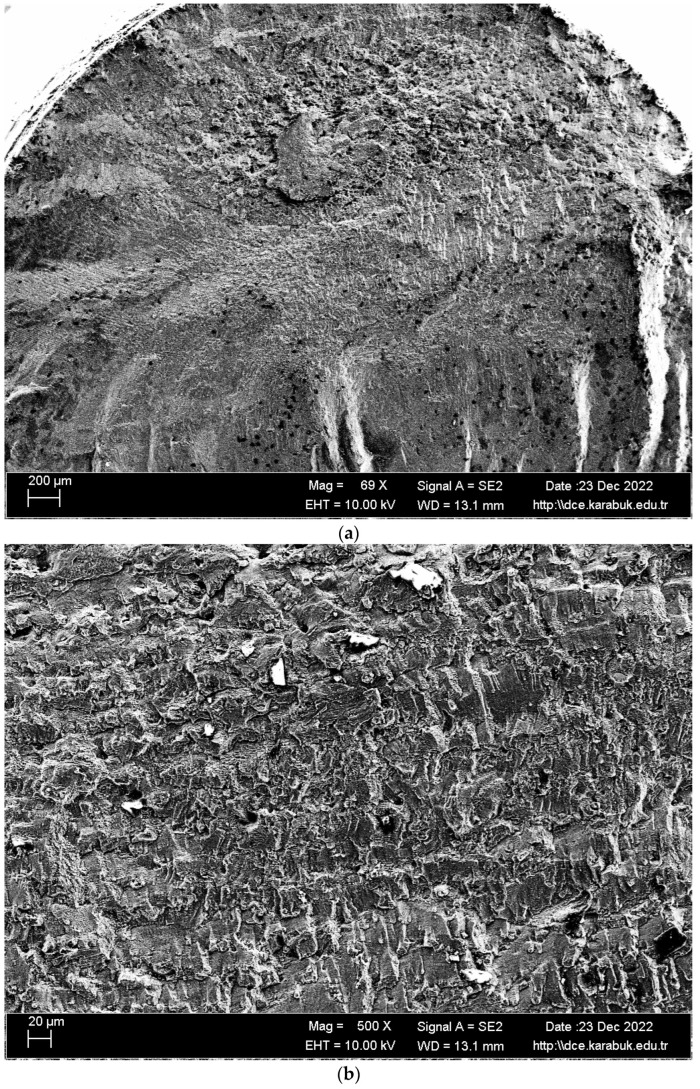
SEM image of the uncorroded Al-5083 fatigue specimen: (**a**) 69×, (**b**) 500×, (**c**) and 1K× magnification.

**Figure 16 materials-16-04996-f016:**
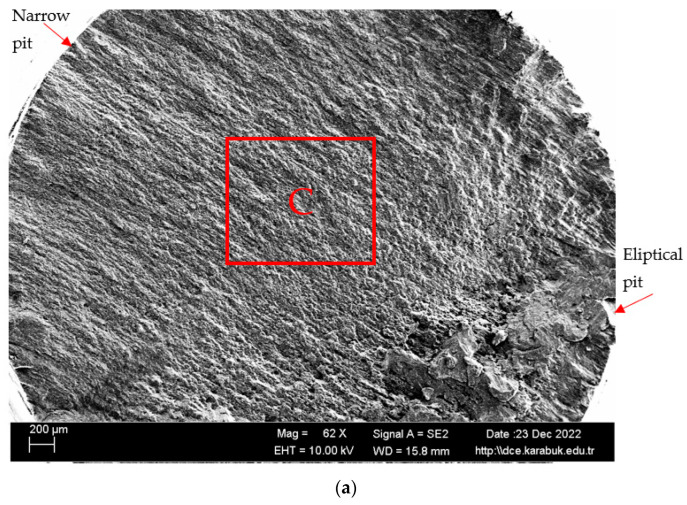
SEM image of the Al-5083 fatigue specimen corroded for 72 h in the 3.5% NaCl solution: (**a**) 62×, (**b**) 500× ((**a**) taken from point C), and (**c**) 1K× magnification.

**Figure 17 materials-16-04996-f017:**
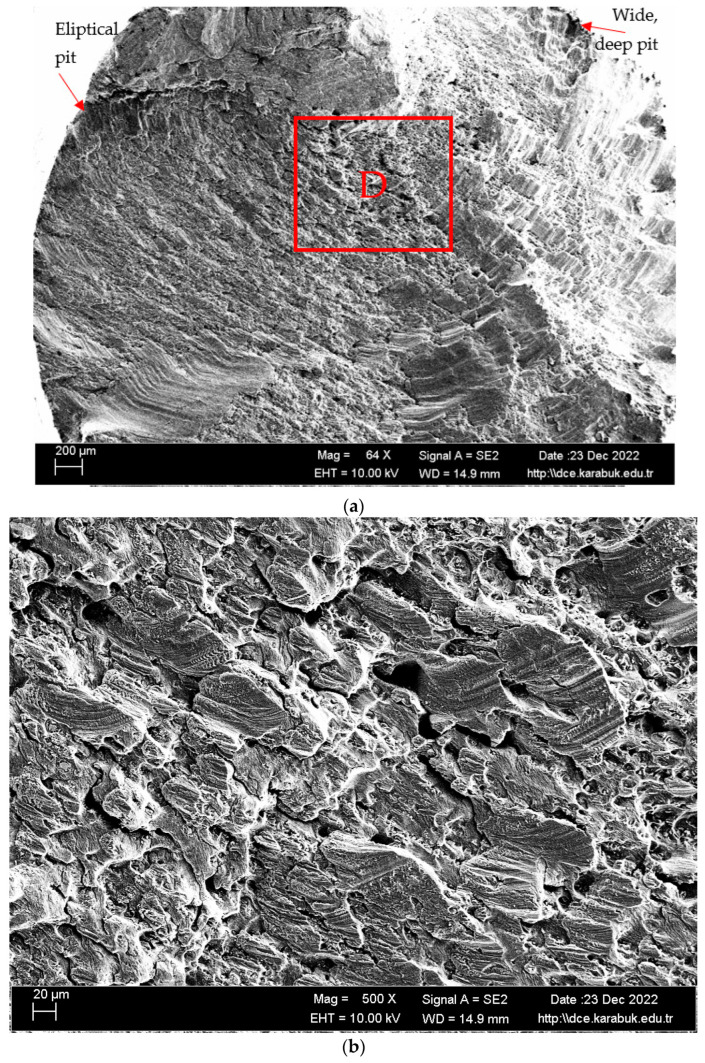
SEM image of Al-5083 fatigue sample etched for 72 h in 3.5% NaCl + 10% HCl solution (**a**) 64× (**b**) 500× ((**a**) taken from point D) (**c**) 1K× magnification.

**Table 1 materials-16-04996-t001:** Typical chemical compositions of the aluminum alloy 5083 [8].

Alloy	Fe	Si	Cu	Mn	Mg	Zn	Cr	Ti	Other	Al
Al5083	0.4	0.4	0.1	0.4–1.0	4.0–4.9	0.25	0.50–0.25	0.15	0.15	Balance

**Table 2 materials-16-04996-t002:** Chemical composition (wt.%) of the hot-rolled Al-5083-H111 sample.

	Mg	Si	S	Cl	Cr	Mn	Fe	Ni	Cu	Zn	Al
Sample 5083	3.6500	0.3486	0.0108	0.0376	0.1218	0.5259	0.3483	0.0126	0.069	0.0345	94.8408

**Table 3 materials-16-04996-t003:** EDX analysis results from Figure 5.

Spectrum	Mg	Al	Si	Cr	Mn	Fe
**1**	5.19	94.81	0.00	0.00	0.00	0.00
**2**	2.06	33.29	63.76	0.28	0.61	0.00
**3**	4.62	93.34	0.92	0.16	0.81	1.06
**4**	5.16	93.68	0.50	0.22	0.45	0.00

**Table 4 materials-16-04996-t004:** EDX analysis results from Figure 11.

Spectrum	O	Na	Mg	Al	Si	Cl	Cr	Mn	Fe
1	1.48	0.53	5.16	92.42	0.33	0.00	0.08	0.00	0.00
2	1.75	0.46	0.29	2.21	94.98	0.17	0.15	0.00	0.00
3	6.57	0.30	4.41	86.48	0.70	0.00	0.40	0.47	0.68
4	7.43	0.44	4.51	86.87	0.41	0.04	0.30	0.00	0.00
5	6.71	0.33	3.91	75.45	12.55	0.01	0.33	0.35	0.34
6	3.96	0.36	0.83	10.37	81.76	0.00	0.26	1.26	1.19

**Table 5 materials-16-04996-t005:** EDX analysis results from Figure 13.

Spectrum	H	O	Na	Mg	Al	Si	Cl	Cr	Mn	Fe
1	0.04	10.77	2.05	4.38	82.13	0.11	0.00	0.00	0.00	0.51
2	76.00	11.48	0.05	0.46	6.16	0.45	0.00	3.67	1.73	0.00
3	3.96	19.09	4.62	3.43	66.13	0.59	0.69	0.58	0.00	0.91
4	0.20	2.20	0.38	4.84	91.97	0.29	0.12	0.00	0.00	0.00
5	2.29	1.79	0.30	4.46	90.02	0.00	0.13	0.00	0.75	0.27
6	1.42	3.84	0.98	4.05	80.41	0.11	8.54	0.12	0.17	0.35
7	16.77	3.43	0.54	1.49	29.39	0.00	47.68	0.00	0.42	0.28

## Data Availability

Not applicable.

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
