# Peer review of "Effect of a 3.5% NaCl−10% HCl Corrosive Environment on the Fatigue Behavior of Hot Rolled Aluminum 5083-H111"

_materials, 2023, doi:10.3390/ma16144996_

Round 1

Reviewer 1 Report

The article (materials-2486888) titled ‘Effect of 3.5% NaCl-10% HCl Corrosive Environment on Fatigue Behavior of the H111 Hot Rolled Aluminium 5083’ reports on the microstructure, hardness, corrosive strength behavior of H111 Al alloy and several conclusions are drawn. The manuscript is well-arranged with fair conclusions. However, I have significant doubts about the novelty of the work. As I see, several articles concerning the same material and test condition are already published in the last decades. Hence, before the manuscript can be published, the following major comments must be addressed in detail with explanations:

ABSTRACT:

1-      Line 12, Please mention strength type- fatigue (torsion), tensile, compression, yield, or any other.

2-      Please give a brief result about the effect on microstructure, and hardness, as well.

3-      Please mention what type of hardness (Vickers, brinell).

4-      Line 18, What are those developed intermetallic phase?

5-      Please refrain from using better, larger, higher, etc. buzz words. Better to include values, or how many times (like twice better, etc.).

INTRODUCTION:

6-      Line 35, please include the low strength value number and relevant citations.

7-      Line 36, what are usually the alloying elements? Are there any apart from Mg? If yes, for what purpose, yield strength, or fatigue strength?

8-      Are there any other standards for these alloys apart from H111, why the choice of H111?

9-      Please mention the novelty of the work. How it is different from already published ones. Is it just about corrosion or microstructure or technology of production is important too?

10-   Line 74, inclusion or the corrosion pit are broadly used general terms. To make it scientific it is better to include what is the size of the pit, what inclusions are detrimental, etc.

11-   Line 84, probably the use of 90% is an overstatement. Better to say a significant or considerably high.

Body:

 12-   Please include how many times tests were repeated (hardness, fracture measurements, etc. if any). Or the values reported are for a single performance?

13-   Tolerance in hardness value should be given. 68.67 HB +- ??

14-   Figure 2, please include the units of dimensions.

15-   Can the image quality be improved? In case, the image scale bar is small (Fig. 4d), I recommend physically drawing a scale bar.

16-   Fig. 4 or 5, which side is the rolling direction? Please mention on the figure with an arrow or something.

17-   As from optical images, I see some defective areas on the as-delivered alloys. How do authors explain the fact that corrosion could be enhanced due to the surface defects (porosity, cracks) on the surface of samples?

18-   Fig. 9- Y axis. Better to just write mg/dm2. “.day” gives confusion. Authors have already said before that the corrosion rate is calculated /day.

19-   Fig. 10a, Fig. 15, Fig. 16, Fig. 17, I would recommend mentioning or writing about the mechanism of material removal. As in, delamination, crack, or pitting corrosion, or where is shallow. And if possible, differentiate between corroded surface, product (as in debris), partially-corroded surface, etc. Mentioning on the figure itself would be better to explain the general corrosion idea. Well-labeled pictures or schematics of mechanisms are impressive and get the work more value.

Conclusions:

20-   Line 394, please mention the size of grains in spite of small or large.

21-   Line 394-395, intermetallic particles or phases? If particles, what size?

22-   Line 397, please give tolerance or error numbers in hardness value.

I would say, better paraphrasing and moderate corrections are required. Please ask a native speaker to help.

Reviewer 2 Report

The pdf file lists the things that need to be improved. The fatigue results are not conclusive. Without these, the work may become scientifically insignificant and of interest to other researchers.

Round 2

Reviewer 1 Report

The manuscript has improved to a good extent. However, I still see some major areas authors should give a thought about or modify:

1- The abstract could be shorter, in case the current word limit goes with the Journal/Editor's choice. It's okay. In general, the abstract is like a conclusion. There is no mention of why the world needs this work. 

2- Figure 8, Y axis. The unit style doesn't impress me. ".day" could be removed. 

3- According to me, two tests are not enough to show results. But it depends, if both the tests have a marginal deviation in values then the average of both can be accepted. However, authors should give some error bars or write somewhere about the noticed deviation in values. Or any significant change noticed during repeatability. At least, in figures and reports of Corrosion rates, the Fatigue tests. It would be better to mention about repeatability of the tests even if marginal difference in rates were noticed.

4- Also, better if values are reported in whole numbers. I don't think numbers after decimal makes any sense here. (Figure 14, 0 hr exposure time).

5- There is no discussion about the "elliptical/wide-deep/narrow pit" in the whole manuscript. But is shown in Fig. 16 a, 17a. I would recommend adding these names to the text and their significance. I mean how do authors decide which is better or worse?

6- Authors must explain the difference in their choice and use of varied terms - Intermetallic phases or intermetallic particles or intermetallic compounds. In the case of particles, please mention their average grain size distribution. 

7- Line 14, Better to replace "ready-made" from "as-delivered/as-obtained".

English seems fine. However, there is a chance of better paraphrasing and making a good flow.

Reviewer 2 Report

At this point, given the changes made as required, I will accept the work in this form. You should take another look at the editing rules. There are some issues that I noticed but didn't report.
